# IGSPAD: Inverting 3D Gaussian Splatting for Pose-agnostic Anomaly Detection

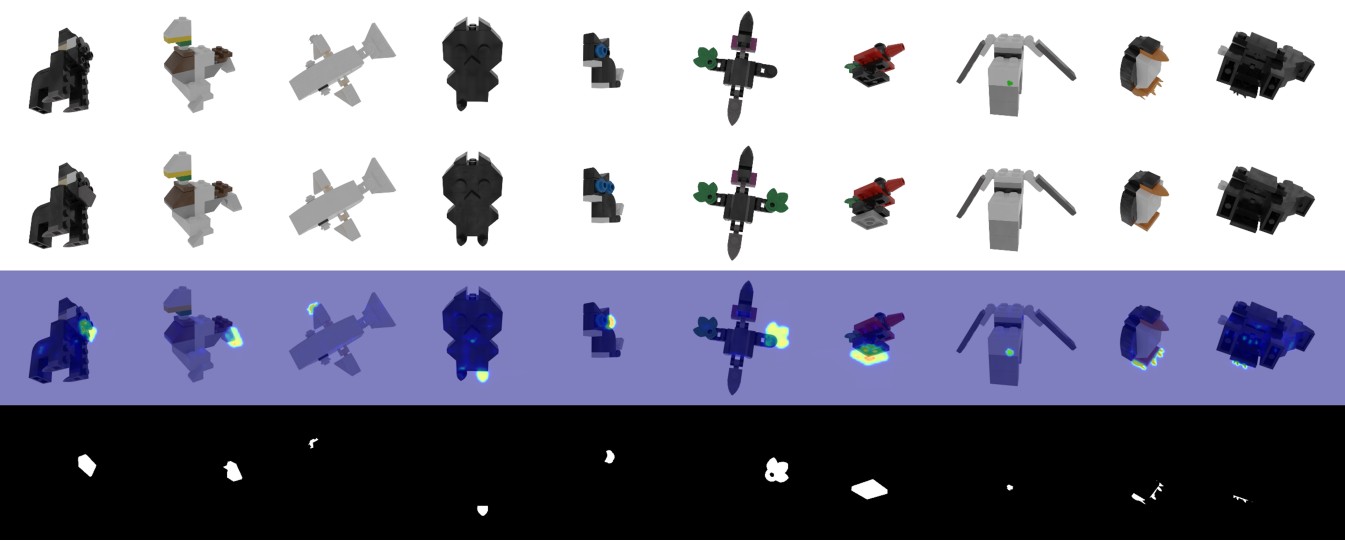

**Figure 1: Visualization of the Pose-agnostic Anomaly Detection result. From top to bottom: the test image, the rendered image, the heat-map of predicted anomaly score map, and the ground truth.**

## ABSTRACT

Pose-agnostic anomaly detection refers to the situation where the pose of test samples is inconsistent with the training dataset, allowing anomalies to appear at any position in any pose. We propose a novel method IGSPAD to address this challenge. Specifically, we employ 3D Gaussian splatting to represent the normal information from the training dataset. To accurately determine the pose of the test sample, we introduce an approach termed Inverting 3D Gaussian Splatting (IGS) to address the challenge of 6D pose estimation for anomalous images. The pose derived from IGS is utilized to render a normal image well-aligned with the test sample. Subsequently, the image encoder of the Segment Anything Model is employed to identify discrepancies between the rendered image and the test sample, predicting the location of anomalies. Experimental results on the MAD dataset demonstrate that the proposed method significantly surpasses the existing state-of-the-art method in terms of precision (from 97.8% to **99.7%** at pixel level and from 90.9% to **98.0%** at image level) and efficiency.

Permission to make digital or hard copies of all or part of this work for personal or classroom use is granted without fee provided that copies are not made or distributed for profit or commercial advantage and that copies bear this notice and the full citation on the first page. Copyrights for components of this work owned by others than the author(s) must be honored. Abstracting with credit is permitted. To copy otherwise, or republish, to post on servers or to redistribute to lists, requires prior specific permission and/or a fee. Request permissions from permissions@acm.org.

*ACM MM, 2024, Melbourne, Australia*

© 2024 Copyright held by the owner/author(s). Publication rights licensed to ACM.
ACM ISBN 978-x-xxxx-xxxx-x/YY/MM
https://doi.org/10.1145/nnnnnnn.nnnnnnn

## CCS CONCEPTS

• **Computing methodologies → Scene anomaly detection**; **Image segmentation**; *3D imaging*; Rasterization.

## KEYWORDS

Anomaly Detection, 3D Gaussian Splatting, Pose Estimation

## 1 INTRODUCTION

Due to the scarcity of anomalous samples, researchers have focused on unsupervised anomaly detection, where only normal samples are present in the training set. This approach has significantly advanced the field of visual anomaly detection. However, current unsupervised anomaly detection methods assume that the test images are well-aligned with the training images, a condition that only represents a subset of real-world scenarios. To further advance the field of visual anomaly detection, we should consider more common scenarios, such as Pose-agnostic Anomaly Detection (PAD). Following the work of Zhou et al. [27], Pose-agnostic Anomaly Detection asserts that there is no precise alignment between training and test set images, with the test set potentially including poses that were never encountered in the training set, thus increasing the complexity of anomaly detection.

The Multi-pose Anomaly Detection (MAD) [27] dataset is the first dataset specifically designed for Pose-agnostic Anomaly Detection (PAD). Testing on the MAD dataset, the efficacy of existing 2D unsupervised anomaly detection methods is significantly reduced. OmniposeAD [27], the first method tailored for the PAD task, greatly improves detection performance compared to conventional

2D unsupervised methods. OmniposeAD utilizes NeRF [15] to represent the normal information of the training set. For each test image, OmniposeAD estimates the camera pose using iNeRF [21] and then renders a normal image based on the estimated pose. The test image and its corresponding rendered image are subsequently processed through feature extraction and compared to localize anomalies area. However, as reported by OmniposeAD, running the complete framework on the MAD dataset requires 10 to 15 hours on a single NVIDIA Tesla A100, indicating low time efficiency. Consequently, there is a pressing need to develop methodologies for the PAD task that not only improve performance but also enhance time efficiency.

Recently, 3D Gaussian Splatting (3DGS) [10] has emerged as a popular method for 3D scene representation, achieving state-of-the-art results in rendering speed and image quality. With its significant advantages in efficiency and quality, 3DGS proves more effective for representing normal information compared to NeRF [15]. Additionally, the differentiability of 3D Gaussians allows for the calculation of gradients of the loss with respect to the camera pose, which enables the estimation of camera poses for test images, addressing the challenge of pose-agnostic in testing.

To this end, we propose our method **I**nverting 3D **G**aussian **S**platting for **P**ose-agnostic **A**nomaly **D**etection (**IGSPAD**). Initially, **IGSPAD** trains 3DGS with normal images from the training set to represent the normal information. Unlike traditional Six-degree-of-freedom pose estimation (6DoF) , estimating the camera pose for anomalous images would be disrupted by the anomalous regions. To address this issue, We specifically designed **I**nverting 3D **G**aussian **S**platting (**IGS**) to estimate the camera poses of the anomalous images. The test image and the normal image rendered under the estimated pose is then processed through a pre-trained feature extractor, and the cosine similarity of the features is calculated to generate the final anomaly detection results.

Our proposed **IGS** involves three principal steps:

**Gradient Calculation**: We explicitly calculate the gradients of the loss with respect to the camera pose using the chain rule.

**Initial Pose Estimation**: Using LightGlue [13], we match the test image against all images in the training set, adopting the camera pose information of the most closely matched training image as the initial pose.

**Matching Points Sampling**: We sample around the points that match between the rendered and test images to minimize the influence of anomalous regions on the camera pose gradients.

Subsequently, we evaluated and compared the efficacy of different feature extractors on anomaly detection performance, ultimately selecting the image encoder of SAM [11] as the feature extractor for our method.

Extensive experiments on the MAD dataset demonstrated the effectiveness of our method. Our approach achieved state-of-the-art performance in Pose-agnostic Anomaly Detection, enhancing the pixel-level AUROC from 97.8% to 99.7% and the image-level AUROC from 90.9% to 98.0%. Additionally, we achieved a pixel-level AP of 75.53%. Our method required only 5 hours on a 4090 GPU, marking at least a twofold improvement in time efficiency over OmniposeAD, which took 10 to 15 hours on an A100 GPU. The main contributions of our method are summarized as follows:

- We introduce a novel method for Pose-agnostic Anomaly Detection called IGSPAD.
- We are the first one who to utilize the 3D Gaussian Splatting model in anomaly detection.
- We propose IGS with MPS, innovatively addressing the challenges of camera pose estimation in the presence of anomalies.
- We conducted a thorough comparison and analysis of various feature extractors, establishing the best practices for maximizing anomaly detection performance.
- Extensive experiments on the MAD dataset demonstrated that our method is effective, significantly enhancing both the performance and time efficiency of anomaly detection and localization.

## 2 RELATED WORK

### 2.1 2D Image Anomaly Detection

Current 2D image anomaly detection tasks are predominantly unsupervised. These tasks involve detecting anomalies under the assumption that the training set consists solely of normal images, while the test set includes both normal and anomalous images. Additionally, it is assumed that the images in both the training and test sets are well-aligned.

*2.1.1 Dataset.* Commonly used datasets for 2D anomaly detection include MVTEC-AD [1], MPDD [8], BTAD [16], and VisA [28].

*2.1.2 Reconstruction-based Methods.* Reconstruction-based methods, such as DRAEM [25], FAVAE [20], and UniAD [22], typically operate under the assumption that models trained solely on normal samples struggle to effectively reconstruct anomalous regions. Consequently, areas in test images showing significant discrepancies before and after reconstruction are more likely to be identified as anomalous.

*2.1.3 Feature Embedding-based.* Feature embedding-based methods, such as PaDiM [4], PatchCore [17], and Fastflow [23], typically employ a pre-trained feature extractor to extract image features and then learn the representation of normal features. Consequently, regions in the test images that deviate significantly from the representation of the normal features are more likely to be identified as anomalous.

*2.1.4 Data Augmentation-based.* Data Augmentation-based methods, such as Defect-GAN [26], AnomalyDiffusion [7] and CAGEN [9], address the challenge of insufficient anomalous samples by simulating or generating anomalies, thereby transforming the unsupervised anomaly detection task into a supervised segmentation or classification task.

### 2.2 Pose-agnostic Anomaly Detection

As introduced by PAD [27], pose-agnostic anomaly detection task operates under the assumption that the images in both training and test sets are not aligned well. The training set includes normal images of various poses along with their pose information, while the test set contains both normal and anomalous images without pose information.

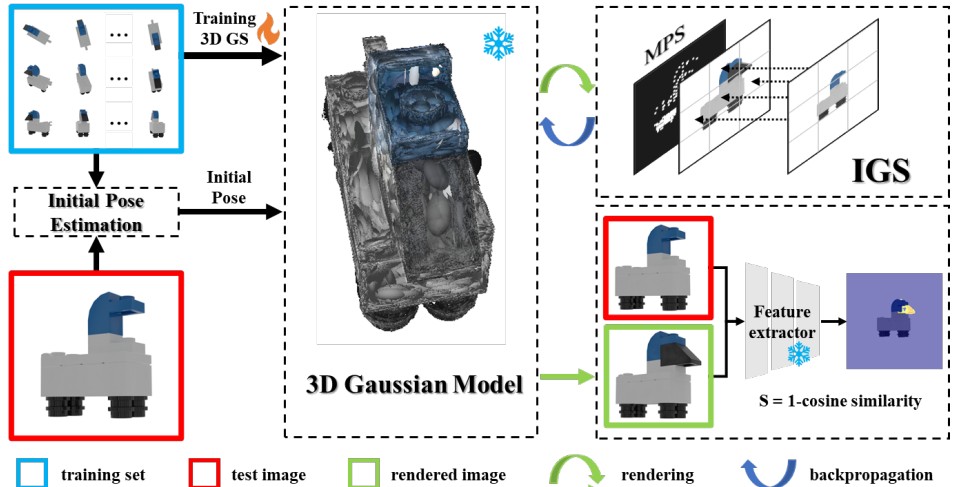

**Figure 2: The framework of our method IGSPAD, which consists of three stages: (i) The 3D Gaussian [10] Model is trained using normal images with pose information to capture the normal information. (ii) Estimate the pose information of the test image through inverting 3D Gaussian Splatting (IGS). (iii) Render the normal image under the estimated pose, calculate the cosine similarity with the test image after feature extraction to obtain the anomaly score.**

PAD introduces the OmniposeAD approach to address pose-agnostic anomaly detection. Initially, OmniposeAD trained a NeRF [15] model using the training set. Given a test image, the camera pose is estimated using iNeRF [21], and this estimated pose information is input into the trained NeRF model to render a normal image that aligns well with the test image. Then comparing these two images and localizing the anomalies.

## 2.3 3D Gaussian Splatting

Recently, 3D Gaussian Splatting (3DGS) [10] has achieved tremendous success in 3D scene representation. As 3D Gaussians are differentiable, gradient descent can be employed to optimize the parameters of 3D Gaussians, achieving a compact 3D representation. Furthermore, the fast differentiable rasterizer for Gaussians enables 3DGS to render high-resolution images rapidly. Those capabilities allows 3DGS to demonstrate significant advantages in high-quality, real-time novel-view synthesis. 3DGS has been successfully applied in various domains, including 3D generation [3] and Simultaneous Localization and Mapping (SLAM) [14].

## 2.4 Sagment Anything

Segment Anything Model (SAM) [11] is a new paradigm for segmentation models, using point prompts and box prompts to indicate the content that the model needs to segment. Trained on a large segmentation dataset of over 1 billion masks, SAM is capable of segmenting any object on a certain image. Works such as Segment Any Anomaly (SAA) [2] are dedicated to applying SAM for anomaly detection.

## 3 APPROACH

In this section, we introduce the proposed method **IGSPAD**. First, we state the definition of pose-agnostic anomaly detection. Subsequently, we detail our anomaly detection approach in two subsections. As shown in Fig. 2, we employ 3DGS [10] to represent the normal information of the training set and then estimate the camera pose of the test image through inverting 3D Gaussian Splatting (**IGS**). To minimize the influence of anomalous regions on gradient backpropagation, we introduce the Matching Points Sampling (**MPS**) method. Finally, we extract features from both the rendered image and the test image, compute the cosine similarity for each pixel, and obtain the anomaly scores.

## 3.1 Problem Definition and Challenges

The concept of pose-agnostic anomaly detection, as defined by PAD [27], assumes that the test samples are not well-aligned with the images in the training set. Specifically, the training set comprises anomaly-free images of objects in the same class but captured in various poses, with attached pose information. In contrast, the poses of the test samples remain unknown and can be arbitrary, potentially including poses not previously encountered in the training set. This discrepancy makes it challenging to find a training image similar enough to a test sample for effective anomaly detection.

## 3.2 Inverting 3D Gaussian Splatting

To address this challenge, we first employ 3D Gaussian Splatting [10] to reconstruct the normal 2D images into a 3D representation, thereby capturing the 3D normal information. As a result, our objective transitions from locating a similar image in the training set to rendering an image from the 3D Gaussian model that aligns accurately with the test image. This task evolves into a 6D pose estimation problem. Drawing on the method similar to iNeRF [21],

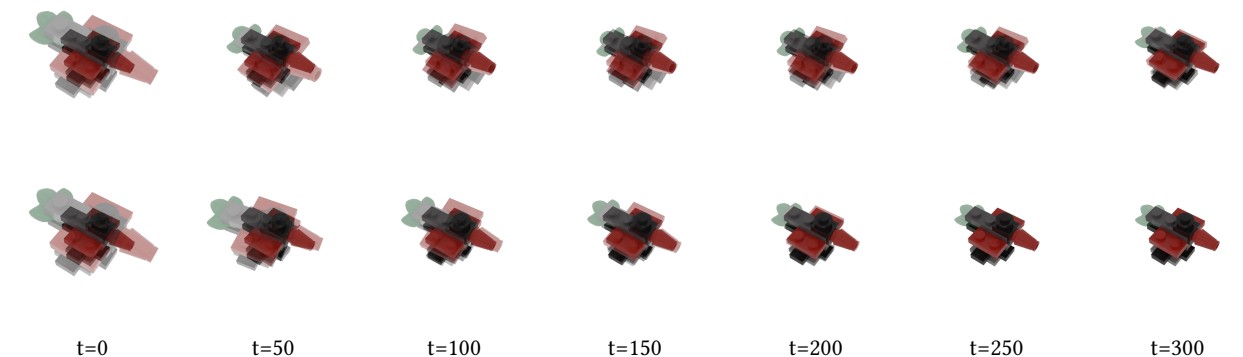

t=0          t=50          t=100          t=150          t=200          t=250          t=300

**Figure 3: The visualization of IGS. The first line is IGS without MPS and the second line is IGS with MPS. IGS without MPS initially expends significant optimization efforts on matching anomalous regions, such as aligning the green tail with the black area. However, MPS can prevent this issue.**

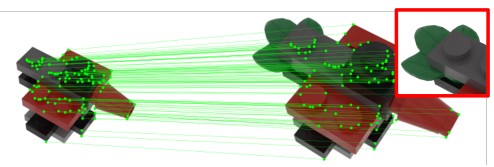

**(a) The visualization of matching between the test image (left) and the rendered image (right). The tail of the bird in the test image is missing.**

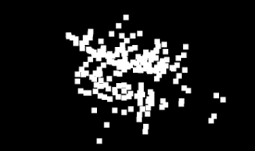
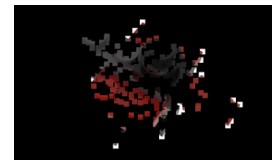

**(b) The weighting map.**          **(c) The sampled image.**

**Figure 4: When backpropagating the camera pose, anomalous regions such as a missing tail can hinder optimization. By using matching point sampling, the impact of these anomalous areas can be eliminated.**

we optimize the camera parameters by computing the gradient with L1 loss respect to the camera pose. Our proposed method Inverting 3D Gaussian Splatting (IGS) comprises three principle steps.

*3.2.1 The gradient with respect to the camera pose.* As the 3D Gaussian is differentiable, we are able to compute gradient of the loss with respect to the camera pose. Gaussian Splatting SLAM [14] provides the analytical Jacobian of the SE(3) camera pose with respect to the 3D Gaussians used in 3DGS. Instead of utilizing Lie algebra as in Gaussian Splatting SLAM, we directly compute the gradient of the translation component $\mathbf{t}$ of the view matrix $\mathbf{T}$ using the chain rule. In 3DGS, a pixel colour $C_p$ can be calculated as follows:

$$C_p = \sum_{i \in N} c_i \alpha_i \prod_{j=1}^{i-1} (1 - \alpha_j) \tag{1}$$

where $c_i$ denotes the learned color of the $i^{th}$ Gaussian, and $\alpha_i$ is related to the 2D Gaussian $N(\boldsymbol{\mu}', \Sigma')$ projected onto the camera coordinate system plane from the $i^{th}$ 3D Gaussian $N(\boldsymbol{\mu}, \Sigma)$ which can be calculated as follows:

$$\boldsymbol{\mu}' = \mathbf{PT}\boldsymbol{\mu}, \Sigma' = \mathbf{JW}\Sigma\mathbf{W}^T\mathbf{J}^T \tag{2}$$

where $\mathbf{P}$ represents the projection operation, $\mathbf{T}$ denotes the view matrix, $\mathbf{J}$ is the Jacobian of the linear approximation of the projective transformation and $\mathbf{W}$ refers to the rotational component of the view matrix $\mathbf{T}$. According to the chain rule, we have:

$$\frac{\partial \mathbf{L}}{\partial \mathbf{t}} = \frac{\partial \mathbf{L}}{\partial C} \frac{\partial C}{\partial \alpha_i} \left( \frac{\partial \alpha_i}{\partial \boldsymbol{\mu}'} \frac{\partial \boldsymbol{\mu}'}{\partial \mathbf{t}} + \frac{\partial \alpha_i}{\partial \Sigma'} \frac{\partial \Sigma'}{\partial \mathbf{t}} \right) = \frac{\partial \mathbf{L}}{\partial \boldsymbol{\mu}'} \frac{\partial \boldsymbol{\mu}'}{\partial \mathbf{t}} \tag{3}$$

According to Eq. (2), $\frac{\partial \boldsymbol{\mu}'}{\partial \mathbf{t}} = \mathbf{J}$, which is easier to calculate. From this, we can explicitly calculate the gradient of the loss with respect to the translation component $\mathbf{t}$ of the view matrix $\mathbf{T}$, and the calculation of the gradient for the rotational component follows the implementation used in Gaussian Splatting SLAM [14].

*3.2.2 Initial Pose Estimation.* From the bad case shown in Fig. 8, a poor initial pose can have a devastating impact on the performance of anomaly detection and it is important to find a good initial pose. We utilize the pose information from the training set, matching the test image with all images in the training set using LightGlue [13]. We assume that the more matched points two images have, the closer their poses are. The pose information from the training image with the highest number of matching points to the test image is used as the initial pose.

*3.2.3 Matching Points Sampling.* Unlike traditional 6D pose estimation problem, pose estimation on anomaly images can be affected by anomalous regions. The losses caused by these anomalous areas can impact the optimization of the camera pose. As Fig. 3 demonstrate, IGS without MPS would focus on compensating for the losses caused by anomalies at an early stage. To address this issue, we propose Matching Points Sampling (MPS).

We first matching the rendered image with the test image using LightGlue [13]. As shown in Fig. 4, the anomalous area can hardly be matched to any point in the rendered image. Then we set the

weights of the pixels within a $k$-pixel radius around the matching points to 1, while the rest are set to 0, resulting in the weighting map. The loss can be caculate as follows:

$$L = \frac{\sum_{i=0}^{n} \sum_{j=0}^{n} |I_r(i,j) - I_t(i,j)| \cdot W(i,j)}{n^2} \quad (4)$$

where $I_r$ and $I_t$ refer to the rendered image and the test image, and $W$ is the weighting map.

As shown in Fig. 4, the sampled image consists mostly of normal regions, the anomalous regions excluded from backpropagation. Fig. 3 demonstrates that IGS with MPS consistently aligns with the normal regions.

## 3.3 Anomaly detection and localization

Given a test image $I_t$, we obtain a pose $\hat{\Theta}$ through IGS and a normal rendered image $I_r$ of $\hat{\Theta}$. Assuming that $I_r$ is well-aligned with $I_t$, anomaly detection and localization then resemble a "spot the difference" task, where the objective is to identify discrepancies between $I_r$ and $I_t$.

Follow Feature-Embedding-based anomaly detection methods like PADIM [4] and PatchCore [17], we obtain the feature map $f_t \in R^{C \times H \times W}$ of $I_t$ and the feature map $f_r \in R^{C \times H \times W}$ of $I_r$. Then the pixel level anomaly score $S_P$ and the image level score $S_I$ between $I_t$ and $I_r$ can be calculate as follows:

$$S_P(i,j) = 1 - \frac{\mathbf{f_t}(:,\mathbf{i},\mathbf{j}) \cdot \mathbf{f_r}(:,\mathbf{i},\mathbf{j})}{\|\mathbf{f_t}(:,\mathbf{i},\mathbf{j})\| \|\mathbf{f_r}(:,\mathbf{i},\mathbf{j})\|} \quad (5)$$

$$S_I = max(AP(S_P)) \quad (6)$$

where $ij$ represents the position of pixels and $AP()$ signifies the operation of average pooling. $S_P$ will be upsampled to the size of the test image $I_t$.

We consider different pretrained models as our feature extractor, ResNet18 [6], WideResNet50 [24], CAIT [18], and the image encoder of SAM [11]. ResNet18, WideResNet50 and CAIT are pre-trained on classification tasks, whereas SAM is pre-trained on segmentation tasks, which is more closely related to the anomaly detection task. Table 1 demonstrates that, compared to other pre-trained models, SAM significantly enhances the capability of anomaly detection, especially the pixel AP metric. Consequently, we ultimately select SAM as our feature extractor.

## 4 EXPERIMENTS

In this section, we first introduce our experimental setup and implementation details. Then, we conduct ablation studies on the individual components of IGSPAD. Finally, we compare our method with other existing Pose-agnostic Anomaly Detection methods.

## 4.1 Experiments Setup

*4.1.1 Dataset.* MAD [27], which means multi-pose anomaly detection dataset, provides 20 categories of LEGO animals toys with diverse shape complexity and color contrasts, constructed using Blender in combination with Ldrew (LEGO parts library). For each category, the training set includes 210 images of different poses along with their corresponding pose information, while the test set comprises 150 to 300 anomalous images and several normal images without pose information. All anomalies are manually generated

**Table 1: Ablation studies with pixel-level AUROC ($AUC_P$), pixel-level AP ($AP_P$), and image-level AUROC ($AUC_I$), grouped by: (i) with or without MPS; (ii) the type of feature extractor; and (iii) the performance of our IGSPAD for reference.**

| Method | MPS | Feature Extractor | $AUC_P$ | $AP_P$ | $AUC_I$ |
|---|---|---|---|---|---|
| OmniposeAD | - | - | 97.8 | - | 90.9 |
| IGSPAD | ✗ | sam_vit_b | 99.60 | 62.44 | 96.63 |
| IGSPAD | ✓ | sam_vit_b | **99.74** | **75.53** | **97.96** |
| IGSPAD | ✓ | resnet18 | 99.00 | 36.58 | 93.70 |
| IGSPAD | ✓ | wide_resnet50_2 | 99.07 | 41.16 | 95.81 |
| IGSPAD | ✓ | cait_m48_448 | 98.57 | 25.23 | 65.65 |
| IGSPAD | ✓ | sam_vit_l | 99.71 | 71.75 | 97.84 |
| IGSPAD | ✓ | sam_vit_h | 99.71 | 72.81 | 97.64 |

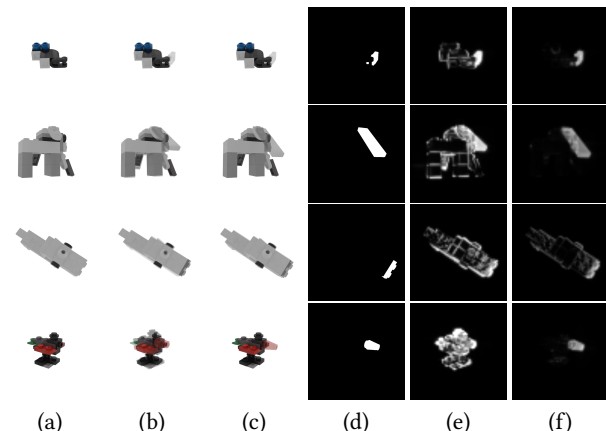

(a)  (b)  (c)  (d)  (e)  (f)

**Figure 5: Comparison of visualization between IGSPAD with and without MSP in terms of rendering and anomaly localization. (a) represents the test image, (b) is the rendered image from IGS without MSP, (c) is the rendered image from IGS with MSP, (d) is the ground truth, (e) shows the anomaly localization results from IGS without MSP, and (f) shows the anomaly localization results from IGS with MSP.**

by professional LEGO players using Photoshop, resulting in sufficiently realistic anomalies. Furthermore, MAD quantifies color contrast and shape complexity for each category in order to assess their influence on the performance of anomaly detection. Detailed information is provided in the appendix. It is important to note that **none** of the data are derived from the real-world scenarios.

*4.1.2 Evaluation Metrics.* Following previous work, we select the Area Under the Receiver Operating Characteristic Curve (AUROC) as our primary evaluation metric. Additionally, we specifically report the pixel-wise average precision (AP) to demonstrate the accuracy of our method in anomaly segmentation.

*4.1.3 Implementation Details.* For each category, the 3D Gaussian model was trained for 100,000 iterations with the image resolution set at 800×800. The IGS process for camera pose estimation involved

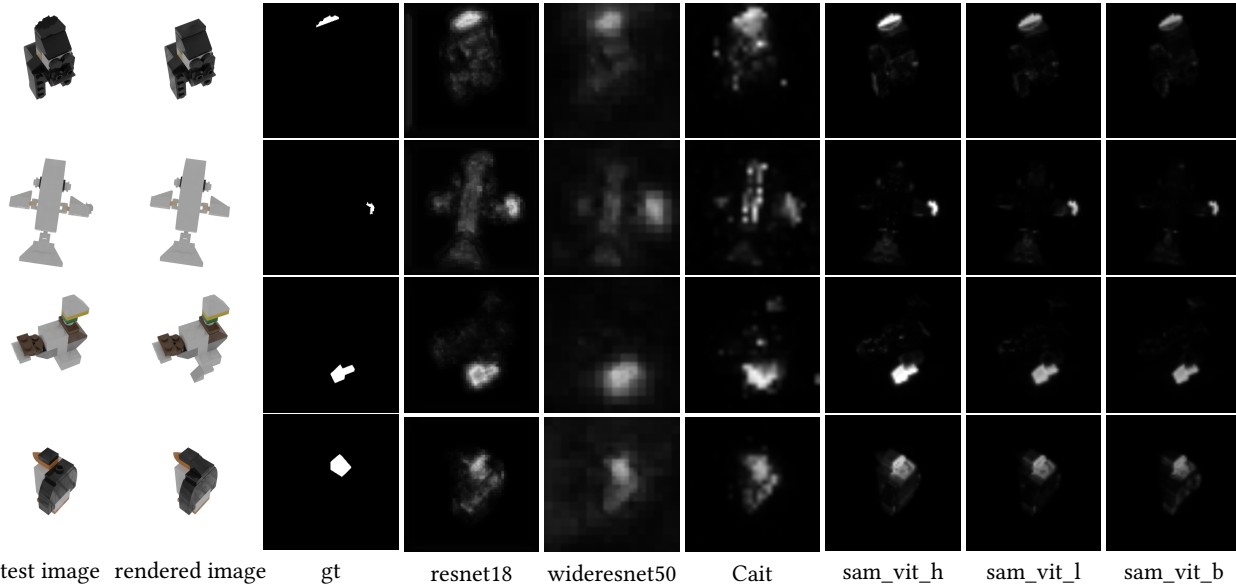

test image    rendered image    gt    resnet18    wideresnet50    Cait    sam_vit_h    sam_vit_l    sam_vit_b

**Figure 6: The visualization of different feature extractors' performance of anomaly detection.**

a total of 300 optimization steps, with the learning rate set to 0.01 and decayed to 0.001 at 200 steps. The feature extractor for IGSPAD has been selected as the image encoder from sam_vit_b. For all 20 categories, training the 3D Gaussian required approximately 160 minutes, pose estimation and image rendering took about 2 hours, and anomaly detection and localization needed roughly 15 minutes, totaling approximately 5 hours. All experiments were conducted on the NVIDIA 4090 GPU.

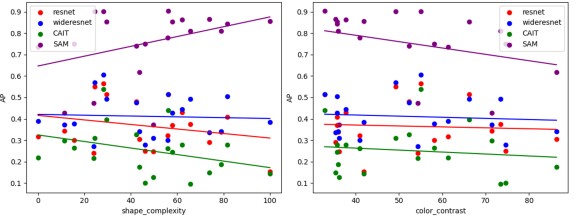

**Figure 7: Correlation between object attributes and anomaly detection performance across various feature extractors**

## 4.2 Ablation Studies

In this subsection, we discuss the effectiveness of MPS and the differences between various feature extractors.

*4.2.1 Matching Points Sampling.* As Table 1 illustrates, the MPS enhances the performance of anomaly detection in all metrics. The augmentation in AP is notably significant. Since most of rendered image without MPS already achieve satisfactory alignment with the test images, the improvement in the AUROC metric is relatively modest.

Fig. 5 showcases instances of suboptimal alignment under the regime without MPS, highlighting that such misalignment can precipitate a significant number of false positives in anomaly localization, thereby exerting a considerable detrimental impact on the AP metric.

Fig. 3 demonstrates the alignment of rendered images and test images in the same iteration step under the conditions of IGS without MPS and IGS with MPS. In the early stages of iteration, IGS without MPS attempts to align the anomalous regions with the normal regions. By the 250th iteration step, IGS with MPS has already aligned, while IGS without MPS still has not aligned by the 300th step.

Compared to simple pose estimation, our method MPS can avoid the influence of anomalous regions on the gradient of camera pose, accelerate the optimization process of camera pose estimation, achieve more accurate camera pose estimation, and significantly improve the performance of anomaly detection.

*4.2.2 Feature Extractor.* As indicated in Table 1, different feature extractors have a significant impact on anomaly detection performance, especially in the AP metric. Not surprisingly, sam, which is pre-trained on segmentation tasks, has a substantial advantage over resnet18, wideresnet50, and CAIT, which are pre-trained on classification tasks.

Fig. 6 presents the visualization results of anomaly localization with different feature extractors. Sam has demonstrated excellent results in anomaly segmentation, showing heightened sensitivity to anomalies as well as robustness to noise in non-anomalous regions. Comparing three different models of SAM with varying amounts of parameters, the model with the fewest parameters surprisingly achieved the best anomaly detection performance. This indicates that it is possible to enhance the time efficiency of anomaly detection without reducing its effectiveness.

Fig. 7 compares the performance of different feature extractors at varying levels of color contrast and shape complexity. Contrary to

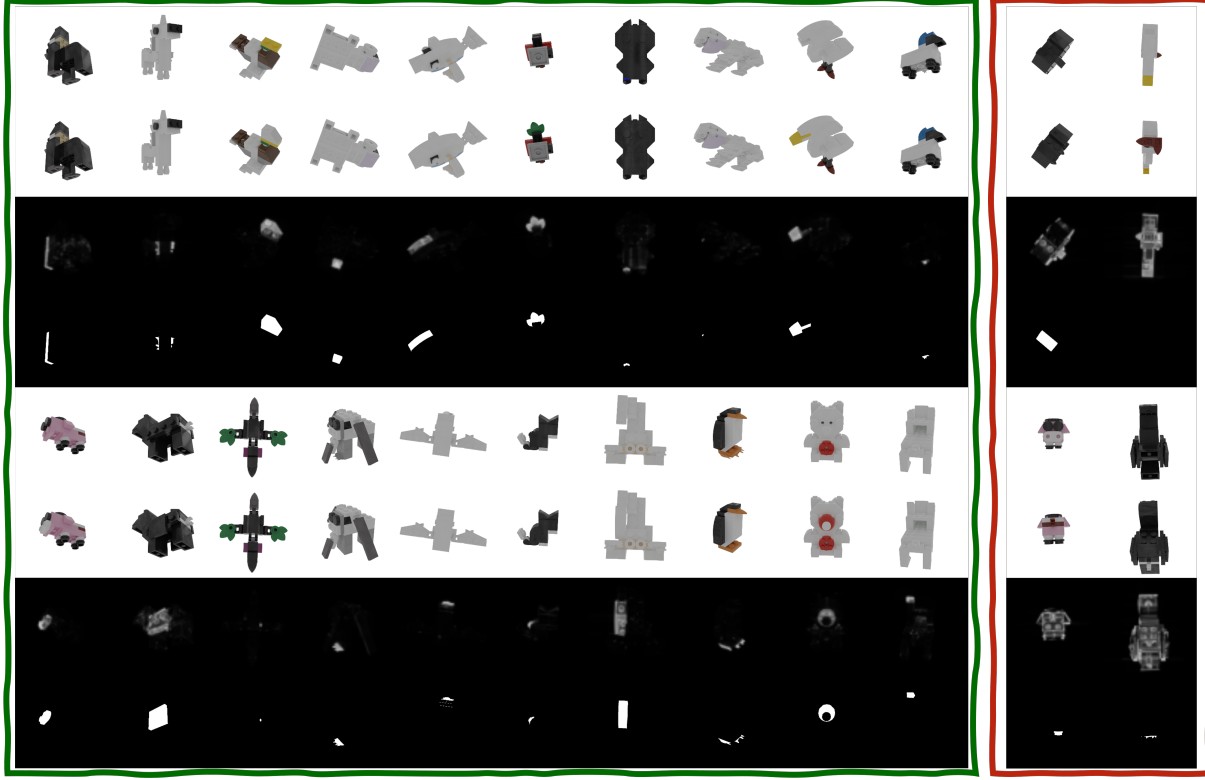

**Figure 8: Qualitative examples. From top to bottom: the test image, the rendered image, the predicted anomaly score map, and the ground truth. The green boxes represent well-predicted cases, while the red boxes denote some bad cases which may be due to the bad initial pose estimation.**

most methods described in PAD [27], our method's performance is negatively correlated with color contrast and positively correlated with structural complexity. Comparing the performance of different feature extractors, there is an overall negative correlation with color contrast for all four types, with ResNet, WideResNet, and CaiT showing more stable performances, which might be more closely associated with IGS. In terms of shape complexity, ResNet, WideResNet, and CaiT demonstrate a negative correlation, while SAM shows a positive correlation, underscoring the advantage of SAM in dealing with complex scenarios. With the increase in shape complexity, the other three feature extractors exhibit a rise in false positives unrelated to anomalies, leading to a reduction in anomaly detection performance.

Overall, SAM as the feature extractor, consistently outperforms the remaining three across all categories, which provides a directive for future training of specialized pre-trained feature extractors dedicated to anomaly detection.

## 4.3 Comparison with State-of-the-art Methods

For a comprehensive comparison, We selected advanced 2D image anomaly detection methods and the SOTA pose-agnostic anomaly detection method OmniposeAD [27]. For the Feature Embedding-based 2D anomaly detection methods, we considered PatchCore [17], STFPM [19], Fastflow [23], CFlow [5] and CFA [12]. For the

Reconstruction-based 2D anomaly detection methods, we selected DRAEM [25], FAVAE [20], and UniAD [22].

Table 2 demonstrates the superiority of our method, which achieves the best performance in almost all categories, except for the image-level AUROC in the Obesobeso. Compared to the current state-of-the-art method, we have achieved an improvement of **1.9** in average pixel-level AUROC and **7.1** in average image-level AUROC. Compared to the method OmniposeAD which also based on rendering and pose estimation, our approach achieves more accurate camera pose estimation and renders higher quality images. Additionally, by applying SAM as a feature extractor, our method significantly improves segmentation accuracy, thereby enhancing its exceptional performance in anomaly detection.

Fig. 8 displays a series of qualitative examples. The results of anomaly prediction are satisfactory most of the time. However, there are still some bad cases where the edges match the test image, but the actual rendering is of its backside or from another angle. These issues are typically due to optimization commencing from an unfavorable initial pose, leading the loss to settle into a local minimum, indicating that a good initial camera pose is extremely important.

**Table 2: Results for anomaly localization / detection with AUROC metric on MAD dataset. The results highlighted in bold represent the best performance.**

| Category | [17] | [19] | [23] | [5] | [12] | [25] | [20] | [22] | [27] | OURS |
|---|---|---|---|---|---|---|---|---|---|---|
| Gorilla | 88.4/66.8 | 93.8/65.3 | 91.4/51.1 | 94.7/69.2 | 91.4/41.8 | 77.7/58.9 | 92.1/46.8 | 93.4/56.6 | 99.5/93.6 | **99.8/96.1** |
| Unicorn | 58.9/92.4 | 89.3/79.6 | 77.9/45.0 | 89.9/82.3 | 85.2/85.6 | 26.0/70.4 | 88.0/68.3 | 86.8/73.0 | 98.2/94.0 | **99.9/99.7** |
| Mallard | 66.1/59.3 | 86.0/42.2 | 85.0/72.1 | 87.3/74.9 | 83.7/36.6 | 47.8/34.5 | 85.3/33.6 | 85.4/70.0 | 97.4/84.7 | **99.9/100.0** |
| Turtle | 77.5/87.0 | 91.0/64.4 | 83.9/67.7 | 90.2/51.0 | 88.7/58.3 | 45.3/18.4 | 89.9/82.8 | 88.9/50.2 | 99.1/95.6 | **99.9/99.1** |
| Whale | 60.9/86.0 | 88.6/64.1 | 86.5/53.2 | 89.2/57.0 | 87.9/77.7 | 55.9/65.8 | 90.1/62.5 | 90.7/75.5 | 98.3/82.5 | **99.9/96.3** |
| Bird | 88.6/82.9 | 90.6/52.4 | 90.4/76.5 | 91.8/75.6 | 92.2/78.4 | 60.3/69.1 | 91.6/73.3 | 91.1/74.7 | 95.7/92.4 | **99.0/99.9** |
| Owl | 86.3/72.9 | 91.8/72.7 | 90.7/58.2 | 94.6/76.5 | 93.9/74.0 | 78.9/67.2 | 96.7/62.5 | 92.8/65.3 | 99.4/88.2 | **99.8/98.5** |
| Sabertooth | 69.4/76.6 | 89.3/56.0 | 88.7/70.5 | 93.3/71.3 | 88.0/64.2 | 26.2/68.6 | 94.5/82.4 | 90.3/61.2 | 98.5/95.7 | **99.9/99.6** |
| Swan | 73.5/75.2 | 90.8/53.6 | 89.5/63.9 | 93.1/67.4 | 95.0/66.7 | 75.9/59.7 | 87.4/50.6 | 90.6/57.5 | 98.8/86.5 | **99.7/97.2** |
| Sheep | 79.9/89.4 | 93.2/56.5 | 91.0/71.4 | 94.3/80.9 | 94.1/86.5 | 70.5/59.5 | 94.3/74.9 | 92.9/70.4 | 97.7/90.1 | **99.4/94.2** |
| Pig | 83.5/85.7 | 94.2/50.6 | 93.6/59.6 | 97.1/72.1 | 95.6/66.7 | 65.6/64.4 | 92.2/52.5 | 94.8/54.6 | 97.7/88.3 | **99.9/98.1** |
| Zalika | 64.9/68.2 | 86.2/53.7 | 84.6/54.9 | 89.4/66.9 | 87.7/52.1 | 66.6/51.7 | 86.4/34.6 | 86.7/50.5 | 99.1/88.2 | **99.5/92.6** |
| Phoenix | 62.4/71.4 | 86.1/56.7 | 85.7/53.4 | 87.3/64.4 | 87.0/65.9 | 38.7/53.1 | 92.4/65.2 | 84.7/55.4 | 99.4/82.3 | **99.9/98.4** |
| Elephant | 56.2/78.6 | 76.8/61.7 | 76.8/61.6 | 72.4/70.1 | 77.8/71.7 | 55.9/62.5 | 72.0/49.1 | 70.7/59.3 | 99.0/92.5 | **99.8/99.7** |
| Parrot | 70.7/78.0 | 84.0/61.1 | 84.0/53.4 | 86.8/67.9 | 83.7/69.8 | 34.4/62.3 | 87.7/46.1 | 85.6/53.4 | 99.5/97.0 | **99.9/99.0** |
| Cat | 85.6/78.7 | 93.7/52.2 | 93.7/51.3 | 94.7/65.8 | 95.0/68.2 | 79.4/61.3 | 94.0/53.2 | 93.8/53.1 | 97.7/84.9 | **99.9/97.7** |
| Scorpion | 79.9/82.1 | 90.7/68.9 | 74.3/51.9 | 91.9/79.5 | 92.2/91.4 | 79.7/83.7 | 88.4/66.9 | 92.2/69.5 | 95.9/91.5 | **98.9/100** |
| Obesobeso | 91.9/89.5 | 94.2/60.8 | 92.9/67.6 | 95.8/80.0 | 96.2/80.6 | 89.2/73.9 | 92.7/58.2 | 93.6/67.7 | 98.0/**97.1** | **99.8**/94.4 |
| Bear | 79.5/84.2 | 90.6/60.7 | 85.0/72.9 | 92.2/81.4 | 90.7/78.7 | 39.2/76.1 | 90.1/52.8 | 90.9/65.1 | 99.3/98.8 | **99.9/99.9** |
| Puppy | 73.3/65.6 | 84.9/56.7 | 80.3/59.5 | 89.6/71.4 | 82.3/53.7 | 45.8/57.4 | 85.6/43.5 | 87.1/55.6 | 98.8/93.5 | **99.9/98.9** |
| Mean | 74.7/78.5 | 89.3/59.5 | 86.1/60.8 | 90.8/71.3 | 89.8/68.2 | 58.0/60.9 | 89.4/58.0 | 89.1/62.2 | 97.8/90.9 | **99.7/98.0** |

## 5 CONCLUSION

In this paper, we propose a novel method IGSPAD for Pose-agnostic Anomaly Detection, which is based on the 3D Gaussian Splatting. We utilize 3DGS to represent the normal information of training set. For the problem of camera pose estimation in anomalous images, we propose the method of Inverting 3D Gaussian Splatting with Matching Points Sampling, which eliminates the impact of anomalous regions on camera pose estimation. We compared various feature extractors and analyzed the impact of features on the performance of anomaly detection. Extensive experiments on the MAD dataset show that, compared to existing state-of-the-art methods, our approach achieved significant improvements. In future work, we plan to further apply our method to anomaly detection in real-world scenarios, to validate whether a training set generated using our approach can effectively detect anomalies in real scenes. Additionally, we will explore how to train a pre-trained feature extractor specifically designed for anomaly detection.

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
