# OpenReview forum: "IGSPAD: Inverting 3D Gaussian Splatting for Pose-agnostic Anomaly Detection"
_acmmm.org/ACMMM/2024/Conference — MM2024 Poster_

### Official Review · Reviewer_ubC8 · 2024-05-25

**Rating:** 3
**Confidence:** 3

**Summary:**

This paper proposes a novel method called IGSPAD (Inverting 3D Gaussian Splatting for Pose-agnostic Anomaly Detection) for pose-agnostic anomaly detection. The key contributions include: 1) Using 3D Gaussian Splatting (3DGS) to represent normal information from the training dataset. 2)Introducing Inverting 3D Gaussian Splatting (IGS) with Matching Points Sampling (MPS) to accurately estimate the pose of test images while avoiding the influence of anomalous regions. 3) Leveraging the image encoder of the Segment Anything Model (SAM) as a feature extractor to identify discrepancies between rendered normal images and test images. 4)Achieving state-of-the-art performance on the MAD dataset, significantly improving precision and efficiency compared to existing methods.

**Strengths:**

1. The proposed IGS with MPS approach provides an innovative solution to the key issue of estimating camera pose in the presence of anomalies.

2. The method achieves impressive state-of-the-art results on the MAD dataset, showing substantial improvements in both precision and efficiency over prior work.

**Limitations:**

1. The writing quality of this paper needs considerable improvement. The authors fail to adequately summarize their main contributions in the introduction and conclusion sections. The abstract does not provide a strong motivation for the proposed work. It appears that the related work section is merely a list of existing studies without a critical analysis.

2. Lack of comprehensive comparisons and discussions with other anomaly detection methods in the field, such as reconstruction-based, feature embedding-based, and data augmentation-based methods. Although comparisons with 2D methods and OmniPoseAD are provided, the paper lacks experimental comparisons with more state-of-the-art methods, especially some recent transformer-based approaches, on both the MAD dataset and other commonly used anomaly detection benchmarks. This makes the experimental section appear less comprehensive.

3. Lack of systematic ablation studies evaluating the effects of different MPS settings. MPS is a key step, but the paper lacks systematic ablation experiments on hyperparameter settings such as the sampling radius and the number of sampling points in MPS. Different sampling settings may significantly influence the robustness to various anomaly patterns (area, distribution, type).

**Suitability:**

2

---

### Official Review · Reviewer_j8Vb · 2024-05-25

**Rating:** 6
**Confidence:** 3

**Summary:**

This paper proposes a novel method, IGSPAD, for pose-agnostic anomaly detection. It uses 3D Gaussian Splatting to represent the normal information of the training set. To address the problem of camera pose estimation in anomalous images, the paper introduces Inverting 3D Gaussian Splatting with Matching Points Sampling, which eliminates the impact of anomalous regions on camera pose estimation. Extensive experiments on the MAD dataset show that the proposed approach achieves new state-of-the-art results.

**Strengths:**

(1)	The paper is, for the most part, well-written and easy to follow.
(2)	The method is novel and effective.
(3)	There is strong experimental evaluation as well as comparisons to baselines.

**Limitations:**

In Table 1, it would be beneficial to include experimental results using the same feature extractor as in reference [27].

**Suitability:**

3

---

### Official Review · Reviewer_S2ZP · 2024-05-25

**Rating:** 5
**Confidence:** 2

**Summary:**

This paper focus on the task of pose-agnostic anomaly detection, where the pose of test samples is inconsistent with the training dataset, allowing anomalies to appear at any position in any pose. To tackle this research problem, this paper proposes an Inverting 3D Gaussian Splatting (IGS) and pose derived from IGS is utilized to render a normal image well-aligned with the test sample. Then image encoder of the Segment Anything Model is employed to identify discrepancies between the rendered image and the test sample, predicting the location of anomalies.

**Strengths:**

1. The primary and most notable strength of this paper is its focus on a novel dataset, Multi-pose Anomaly Detection (MAD), which addresses a new research problem. The MAD dataset is the first specifically designed for Pose-agnostic Anomaly Detection (PAD). Consequently, this work represents a pioneering effort in this area.

2. This paper heavily relies on the recent method of 3D Gaussian Splatting, which is popular and effective for 3D scene representation. It introduces a novel variant, Inverting 3D Gaussian Splatting, which effectively generates the final anomaly detection results.

3. The experiments conducted on the newly released dataset are comprehensive, and the results are outstanding.

**Limitations:**

For a very new task, without established baseline methods (this work would be the first one), it is challenging to identify the limitations of this paper by comparing it with previous methods. Despite this, I find the article's approach simple, logical and effective, with good visual presentation and writing quality, leading me to lean towards acceptance.

There is an error in the citation: PAD is published in NeurIPS 2023, not 2024 (line 1004).

**Suitability:**

2

---

### Meta-Review · Area_Chair_EUsc · 2024-07-01

**Recommendation:** Accept (Poster)
**Confidence:** 5

**Metareview:**

Despite on borderline reject from one of the reviewers, the AC concurs with the two positive reviewers and recommend for publication of this paper. However, the authors should consider to release their code to the public as it is an industry standard nowadays. The authors should address all issues raised by the reviewers in their final version of the paper.